Auditory processing ability in Thai native speakers with the Gaps-In-Noise (GIN) test

Jeanbunjongkij Supakarn 1
Suphinnapong Pawichaya 1 2 3
Limkitisupasin Patcharaorn 1 4
Roongthumskul Yuttana 5
Utoomprurkporn Nattawan nattawan.u@chula.ac.th 1
1 Department of Otolaryngology, Faculty of Medicine, Chulalongkorn University , Bangkok , Thailand
2 University College London Ear Institute , London , United Kingdom
3 Global Health Research Center, Faculty of Medicine, Chiang Mai University , Chiangmai , Thailand
4 School of Otolaryngology, Institute of Medicine, Suranaree University of Technology , Nakorn Ratchasima , Thailand
5 Faculty of Science, Chulalongkorn University , Bangkok , Thailand
Reser David
Electronic publication date: 2025 Nov 6
Publication date: 2025
Volume: 13
Electronic Location ID: e20207
Received 2025 Mar 5; Accepted 2025 Sep 18
Copyright: ©2025 Jeanbunjongkij et al.
Copyright year: 2025
Copyright holder: Jeanbunjongkij et al.
License: This is an open access article distributed under the terms of the Creative Commons Attribution License, which permits unrestricted use, distribution, reproduction and adaptation in any medium and for any purpose provided that it is properly attributed. For attribution, the original author(s), title, publication source (PeerJ) and either DOI or URL of the article must be cited.
License URL: https://creativecommons.org/licenses/by/4.0/

Keywords: Gaps-In-Noise, Auditory process disorder, Auditory norms, Tonal language user

Funding: Thailand Science Research and Innovation (TSRI), Thailand, through Program Management Unit for Competitiveness (PMU C) C23F670098 C23F670098 Royal Academy of Engineering, UK under scheme “transforming system through partnership,” Thai- UK British Council Thailand and Ministry of Higher Education, Science, Research, and Innovation, Ratchadapiseksompotch Fund, Faculty of Medicine, Chulalongkorn University GA66/15,RA-MF14/66 RA-MF02/66,RA-MF-28/68 This work has been funded by Thailand Science Research and Innovation (TSRI), Thailand, through Program Management Unit for Competitiveness (PMU C), contract number C23F670098 and C23F670098 in partnership with Royal Academy of Engineering, UK under scheme “transforming system through partnership,” Thai- UK world class university catalyst grant year 3–4 in partnership between British Council Thailand and Ministry of Higher Education, Science, Research, and Innovation, Ratchadapiseksompotch Fund, Faculty of Medicine, Chulalongkorn University [grant number GA66/15,RA-MF14/66, RA-MF02/66,RA-MF-28/68]. The funders had no role in study design, data collection and analysis, decision to publish, or preparation of the manuscript.

==============================
Introduction

Auditory processing disorder (APD) arises from dysfunctions within the central auditory nervous system (CANS). One key tool for assessing temporal auditory processing capabilities is the Gaps-In-Noise (GIN) test. While the GIN test serves as a critical tool, its accuracy may be influenced by linguistic factors. This research aims to establish preliminary normative data for the GIN test among young adult native Thai speakers and to compare potential performance differences across ears and various linguistic backgrounds.

Methods

This study included 52 native Thai speakers aged 18 to 25, all of whom had normal hearing confirmed through pure-tone audiometry, to examine auditory processing using the GIN test. The testing was conducted in an anechoic chamber to ensure standardized conditions. The results were analyzed based on the approximate gap-detection threshold (A.th.) value and the overall percent correct (OPC) value metrics.

Results

The study revealed that, among young adult native Thai speakers, the A.th. values were 5.12–0.81 milliseconds (ms.) in the right ear, 5.08–0.81 ms. in the left ear, and 5.04–0.68 ms. for both ears. The corresponding averages of OPC values were 69.94–7.66% for the right ear, 71.37–7.21% for the left ear, and 71.35–6.72% for both ears, respectively. While no significant differences were observed in the threshold between ears, a notable discrepancy in response accuracy was noted, with the right ear showing lower accuracy than the left ear. Significant differences were also found when compared to English, Korean, and Portuguese language users, but not when compared to Polish language users.

Conclusion

The study revealed comparable temporal processing abilities among young adult native Thai speakers, as indicated by similar A.th. values across both ears. However, differences in the OPC value responses suggest potential asymmetries in auditory processing mechanisms. Additionally, linguistic factors appeared to influence the outcomes, as evidenced by variations in the A.th. values between Thai speakers and individuals from other linguistic backgrounds. These findings underscore the importance of incorporating language-specific norms when assessing the APD.

Introduction

Auditory processing disorder (APD) is characterized by a diminished ability to interpret auditory information, affecting both verbal and non-verbal inputs (American Speech-Language-Hearing Association, 2005). Individuals with APD may experience deficits in sound localization, auditory discrimination, pattern recognition, temporal processing, and have difficulty functioning in acoustically challenging environments (Moore et al., 2018). Despite normal audiometric results, they often exhibit listening difficulties and behaviors similar to those observed in individuals with hearing loss (American Academy of Audiology, 2010).

Individuals with APD often face challenges in listening and may exhibit behaviors commonly associated with hearing loss, despite having normal hearing test results (American Academy of Audiology, 2010). The root cause of APD is linked to dysfunctions in the neural system, potentially involving damage to both the afferent and efferent pathways within the central auditory nervous system (CANS) (Moore et al., 2018). Central auditory processing encompasses a range of mechanisms aimed to preserve, refine, analyze, modify, organize, and interpret auditory information from the peripheral auditory system. These sophisticated processes form the foundation for essential auditory skills, including auditory discrimination, temporal processing, and binaural processing (American Academy of Audiology, 2010). Auditory temporal processing pertains to the perception of the temporal aspects of a sound, including the capacity to recognize and interpret alterations in the length of sound signals within a particular or defined time frame (Musiek et al., 2005).

The Gaps-In-Noise (GIN) test is a widely used tool for assessing temporal resolution, defined as the ability to detect rapid changes in the auditory signal envelope over time. This test has demonstrated 96.3% sensitivity and 94.4% specificity in detecting lesions in the central auditory nervous system (Paulovicks, 2008). During the test, a 6-second auditory stimulus is presented, intermittently interrupted by 0-3 silent intervals, or “gaps”, which range in duration from 2 to 20 milliseconds (ms.). Participants are required to press a button upon detecting these gaps within the sound. The test yields two primary metrics: the approximate gap-detection threshold (A.th.) value and the overall percent correct (OPC) (Musiek et al., 2005). The A.th. value refers to the length of the gap in the sound that the participant can correctly identify at least 66.67% of the time or four out of six trials. The OPC value refers to the participant’s total accuracy in identifying gaps within the auditory stimulus (sound) (Musiek et al., 2005).

The effectiveness of the GIN test is influenced by the linguistic characteristics of the individuals being assessed (Musiek et al., 2005; Kumar, Ameenudin & Sangamanatha, 2012). For instance, Thai, a tonal language, is known for its slowest speech rate compared to other languages. Some studies have shown that speakers of tonal languages often possess enhanced auditory discrimination abilities when compared to speakers of non-tonal languages (Bidelman, Gandour & Krishnan, 2011). Due to these linguistic complexities, it has been suggested that country-specific standards could be developed for the assessment of APD (Dillon et al., 2012). While standardized assessments and normative data for diagnosing APD exist in English speaking countries, many other nations, including Thailand, lack tailored assessments and normative benchmarks in their native languages. Literature reviews reveal significant variations in normative data across different countries. For instance, in England, the A.th. value in the right ear was 4.9 ms. with an OPC value of 70.3%, while the left ear showed an A.th value of 4.8 ms. with an OPC value of 70.2% (Musiek et al., 2005). In South Korea, the right ear exhibited an A.th. value of 4.95 ms. with an OPC value of 72.08%, and the left ear showed an A.th. value was 5.05 ms., with an OPC value of 72.13% (Choi, Kim-Lee & Jang, 2013). In Brazil, an A.th. value was 4.19 ms. with an OPC value of 78.89% for both ears (Giannela Samelli & Schochat, 2008). In Poland, the A.th. values were 5.4 ms. in the right and 5.3 ms. in the left ear, although no OPC data were provided (Majak et al., 2015).

This research focuses on three primary objectives:

(1) to establish normative data for the GIN test in individuals who are native Thai speakers;

(2) to compare the normative standards of the GIN test across the left ear, right ear, and both ears;

(3) to compare the normative values of the GIN test between Thai native speakers and individuals from other linguistic backgrounds.

Methods

Ethical approval

The study received ethical approval from the Research Ethics Review Committee for research participants, Group 1, Chulalongkorn University. COA No.008/2022, Study number 198.1/64.

Participants inclusion criteria

To ensure normal hearing norms, this study included only young adult native Thai speakers with normal hearing, aged between 18 and 25 years. The participants were required to undergo a normal otoneurological examination and to have a normal hearing threshold of 20 decibel (dB) hearing level or better across frequencies ranging from 250 to 8,000 Hz in both ears. Additionally, they needed to demonstrate a speech recognition threshold (SRT) test, difference of 10 dB or less from the pure-tone average (PTA) test at 500, 1,000, 2,000, and 4,000 Hz, along with a speech discrimination score (SDS) test exceeding 90%. Exclusion criteria included a history of neurological disorders or head trauma, as well as self-reported difficulties such as hearing deterioration, challenges in noisy environments, or learning and language disabilities.

Participants

This study enrolled 52 participants. All participants provided voluntary informed written consent before their inclusion in the study. Subsequently, they underwent a series of procedures, including history-taking, a comprehensive otoneurological examination, and a pure-tone audiogram. The pure-tone audiogram was conducted in an anechoic chamber within the acoustic laboratory at the Mahamakut Building, Faculty of Science, Chulalongkorn University. This assessment covered frequencies of 250, 500, 1,000, 2,000, 4,000, 6,000, and 8,000 Hz for the air conduction test, and 500, 1,000, 2,000, 3,000, and 4,000 Hz for the bone conduction test, followed by the SRT and SDS tests. Participants who met the inclusion criteria were recruited for the study and subsequently underwent the GIN test. The GIN test was recorded on a CD sourced from Auditec, Inc. (St. Louis, MO, USA), and was administered in an anechoic chamber within the acoustic laboratory, consistent with the conditions for pure-tone audiometry. The audiometer was calibrated to deliver sound levels set at 25–30 dB above the SRT, ensuring that the participants could hear the test stimuli clearly and comfortably during both monaural and binaural testing.

The GIN test

The GIN test consists of four lists, each designed to present a sound for 6 s, intermittently interrupted by silent intervals, or “gaps”, occurring 0-3 times within that duration. These silent intervals, known as “gaps,” were varied at durations of 2, 3, 4, 5, 6, 8, 10, 12, 15, and 20 ms. A counterbalanced rotational design was employed, in which 52 participants were each assigned to complete three out of four examination lists, following a rotating sequence. Each participant completed a unique combination of three tests in a cyclic pattern. For example, Participant 1 completed exams 1, 2, and 3; Participant 2 completed exams 2, 3, and 4; Participant 3 completed exams 3, 4, and 1; and so forth. Participants were instructed to press a button upon detecting these gaps, with correct interpretation defined as a response occurring within 1 s of the actual gap time. To minimize fatigue and maintain performance consistency, a 5-minute rest period was provided between each test session. The assessment yields two key metrics: the A.th. value and the OPC value. The A.th. value represents the gap duration that the participant can accurately identify at least 66.67% of the time or four out of six trials. The OPC value measures the proportion of correct button presses when detecting the gaps in the auditory stimulus.

Statistical and data analysis

Based on the literature review and related research by Musiek et al. (2005) on “GIN Test Performance in Subjects with Confirmed Central Auditory Nervous System Involvement,” the study found that administering the GIN test using the A.th. value at the 5 ms. cutoff point yielded a specificity of 84.0%. This finding can be utilized to calculate the sample size as follows (Hajian-Tilaki, 2014): n=Zα22SP ^1−SP ^d2

where: n = sample size

Z∝2 = standard Type I error, with ∝ = 0.05, Z(0.975) = 1.96

SP ^ = specificity of A.th. at the 5 ms. cutoff point equals 84.0%

d = acceptable margin of error, set at 10.0%

Consequently, this study included a total sample size of 52 participants.

The normative values of the GIN test, based on the A.th. and the OPC value responses in individuals among the Thai native language users, are presented as means (M) and standard deviation (SD). Data normality for both the A.th. and OPC responses was evaluated using the Shapiro–Wilk test. Since the results revealed non-normal distributions, non-parametric statistical methods were subsequently employed for all analyses. The comparison of normative data of the GIN test between the left ear, right ear, and both ears is conducted using the Wilcoxon Signed-Rank test. Additionally, the comparison of normative values of the GIN test between individuals who are Thai native language users and those who are other native language users is performed using the One sample Wilcoxon Signed Rank test. The relationship between the A.th. and the OPC responses is analyzed using Spearman correlation (ρ) and presented using scatter plots. Data analysis for this study was conducted using the IBM SPSS Statistics for Windows Version 26.0 (IBM Corp., Armonk, New York, USA, 2019). All tests were two-tailed, and statistical significance was set at the 0.05 level (α = 0.05) with adjusted for multiple comparisons using Bonferroni correction.

Results

A total of 52 participants, aged 18 to 24 years, with a mean age of 20.16 years, participated in the evaluation. The group included 30 males and 22 females. The PTA audiogram testing revealed average thresholds of 7.35 dB in the right ear and 6.90 dB in the left ear. The SRT result was at 10.25 dB in the right ear and 9.5 dB in the left ear, while the SDS was 99.8% in both ears. The average A.th value for Thai native speakers with normal hearing is 5.12 ± 0.81 ms. in the right ear, 5.08 ± 0.81 ms. in the left ear, and 5.04 ± 0.68 ms. in both ears. The OPC value is 69.94 ± 7.66% in the right ear, 71.37 ± 7.21% in the left ear, and 71.35 ± 6.72% when testing both ears. These findings are summarized in Table 1.

The A.th value between the right ear and the left ear, right ear and both ears, and left ear and both ears showed no statistically significant differences (P > 0.05/3). However, the OPC value in the right ear was significantly lower than in the left ear at a significance level of 0.05 (MD −1.44; effect size (r)−0.35; P = 0.012). However, both the right ear and the left ear have no statically significant difference with that of both ears (P = 0.027 and, P = 0.991, respectively). This is shown in Table 2 and Fig. 1.

In Fig. 2 there is a statistically significant negative correlation between the approximate gap-detection threshold and the OPC in the right ear (ρ =  − 0.884), left ear (ρ =  − 0.851), and both ears (ρ =  − 0.840) at a significance level of 0.05, as depicted in Fig. 3

When comparing the A.th values between individuals who are Thai native language users and those who primarily use other languages with normal hearing, it was found that the A.th in Thai language users was significantly higher than in English language users in the right and left ear (P = 0.002 and P = 0.005, respectively). In addition, the A.th in Thai language users was significantly higher than in Korean language users in the right ear (P = 0.002), but there was no statistically significant difference in the left ear (P = 0.196). Moreover, the A.th in Thai language users was significantly higher than in Portuguese language users bilaterally (P < 0.001). Conversely, there was no statistically significant difference between Thai and Polish language users in either the right ear (P = 0.272) or the left ear (P = 0.196). These findings are illustrated in Table 3.

Table 1 Values of the approximate gap-detection threshold (A.th.) and the overall percentage correct (OPC) of the right ear, left ear, and both ears.

Ear (s)	A.th. (msec)	OPC (%)	
	Mean	SD	Range	Mean	SD	Range	
Right	5.12	0.81	4.00–6.00	69.94	7.66	58.30–85.00	
Left	5.08	0.81	3.00–6.00	71.37	7.21	58.30–90.00	
Both	5.04	0.68	4.00–6.00	71.35	6.72	60.00–86.70	
Notes.

SD, Standard deviation.

Table 2 The comparison of the approximate gap-detection threshold (A.th) values and the overall percent correct (OPC) values between the right ear and left ear and both ears.

Paired	Right ear vs Left ear	Right ear vs Both ear	Left ear vs
Both ear	
A.th. (ms.)	Mean difference	0.04	0.08	0.04	
95% CI	−0.11, 0.18	−0.10, 0.25	−0.13, 0.20	
P-value	0.598	0.376	0.642	
OPC (%)	Mean difference	0.08	−1.41	0.03	
95% CI	−0.10, 0.25	−2.66, −0.16	−1.13, 1.20	
P-value	0.017*	0.028*	0.956	
Notes.

Data were analyzed with a Paired t-test.

* Statistically significant at the 0.05 level.

Figure 1 Mean approximate gap-detection threshold value (A.th.) (A) and the mean of overall percent correct (OPC) (B).

Figure 2 Approximate gap-detection threshold (A. th.) of each subject.

Figure 3 Scatter plot showing the correlation between approximate gap-detection threshold (A.th) values and overall percent correct (OPC) values.

Table 3 The comparison of the approximate gap-detection threshold (A.th) value between Thai native speakers and other languages.

Language	Articles	Ear (s)	 A.th. (ms.)	Mean difference (95% CI)	P -value	
Thai	This study	Right	5.12 ± 0.81	n/a		
Left	5.08 ± 0.81	n/a		
Both	5.04 ± 0.68	n/a		
English	Musiek et al. (2005)	Right	4.9a	0.22
(−0.01, 0.44)	0.06	
	Left	4.8a	0.27
(0.05, 0.50)	0.017*	
 	Both	n/a	n/a		
Polish	Majak et al. (2015)	Right	5.4a	−0.28
(−0.51, −0.06)	0.014*	
		Left	5.3a	−0.22
(−0.45, 0.00)	0.053	
 	 	Both	n/a	n/a		
Korean	Choi, Kim-Lee & Jang (2013)	Right	4.95 ± 1.18	0.17
(−0.06, 0.39)	0.146	
		Left	5.05 ± 1.15	0.03
(−0.20, 0.25)	0.812	
 	 	Both	n/a	n/a	n/a	
Portuguese	Giannela Samelli & Schochat (2008)	Right	n/a	n/a	n/a	
		Left	n/a	n/a	n/a	
 	 	Both	4.19a	0.85
(0.66, 1.04)	<0.001*	
Notes.

Data were analyzed with One-sample t-test.

* Statistically significant at the 0.05 level.

a SD not reported in original study.

Discussion

The APD poses a substantial challenge in clinical practice, frequently manifesting as difficulties in the interpretation of auditory signals, despite individuals presenting with normal hearing thresholds. Our study aimed to investigate auditory processing ability in Thai native language users, focusing on establishing normative values for the GIN test, comparing these values between ears, and examining the potential for auditory differences between Thai native language users and individuals from other linguistic backgrounds. The results of our study offer valuable insights into auditory processing abilities among Thai native language users. Temporal resolution was assessed using two key metrics: A.th. value and the OPC value responses. The findings indicate that Thai native language users had the A.th. values of 5.12 ± 0.81 ms. in the right ear, 5.08 ± 0.81 ms. in the left ear, and 5.04 ± 0.68 ms. when tested bilaterally. Additionally, the OPC values were 69.94 ± 7.66% in the right ear, 71.37 ± 7.21% in the left ear, and 71.35 ± 6.72% in both ears. These normative data provide a baseline for evaluating temporal processing abilities in Thai native language users and can serve as a reference for future assessments in clinical and research settings.

Interestingly, our study revealed no significant differences in the A.th. values between the right and left ears or between unilateral and bilateral testing conditions. However, a statistically significant difference was observed in the OPC responses, with the right ear showing slightly lower accuracy compared to the left ear and bilateral testing. While this finding may reflect subtle asymmetries in auditory processing, caution is warranted in its interpretation. Small p-values can sometimes result from minimal effect sizes, and such differences may not always reflect meaningful clinical or functional disparities. Factors such as individual variability, neural asymmetry, or handedness could contribute, but further investigation is needed to determine whether these effects are robust and generalizable. These results underscore the complexity of auditory perception and suggest that even in the presence of similar threshold values, small variations in response accuracy may occur across conditions

Furthermore, our study compared auditory processing abilities between Thai native language users and individuals from other language backgrounds. We found that Thai native language users exhibited higher A.th. value compared to English, Korean and Portuguese language users, suggesting potential differences in auditory processing between tonal and non-tonal languages. This finding may reflect the unique acoustic features and perceptual demands associated with tonal languages, which may influence temporal processing abilities.

Additionally, Thai language users demonstrated lower A.th. value compared to Polish language users, indicating distinct auditory processing characteristics across different language groups. These results underscore the importance of considering linguistic factors when assessing auditory processing abilities, as they may shape the way individuals perceive and process auditory stimuli. The linguistic background of the participants could influence how effectively they detect temporal gaps due to differences in phonological processing and auditory experience. At the same time, methodological variations—such as differences in test presentation or order method, kind of equipment, stimulus intensity, background noise control, or calibration could affect gap perception. And participant selection method—can amplify or obscure those linguistic effects, leading to variability in results across studies.

The significant negative correlations observed between the A.th and the OPC responses in the right ear (ρ =  − 0.884), left ear (ρ =  − 0.851), and both ears (ρ =  − 0.840) suggest a strong inverse relationship: as gap-detection thresholds decrease (indicating better temporal resolution), the accuracy of responses increases. This pattern aligns well with theoretical expectations, thereby supporting the construct validity of the test (Streiner, Norman & Cairney, 2015). Construct validity refers to how well a test measures the concept it is intended to assess. In this case, gap-detection ability is a key indicator of auditory temporal resolution. The strong negative correlation between threshold values and performance implies that the test is successfully capturing individual differences in this construct. According to Streiner, Norman & Cairney (2015), evidence of such meaningful and theoretically consistent relationships is essential in establishing validity in clinical psychometric assessments. Moreover, the consistency of these correlations across both ears indicates reliability—that is, the test provides stable and replicable results across different but comparable conditions (right vs. left vs. both ears). This contributes to the internal consistency of the test and suggests that it performs robustly across various auditory input pathways.

However, it is worth noting that strong correlations alone do not confirm all aspects of reliability (e.g., test–retest reliability over time), nor do they fully account for potential sources of error variance, especially in clinical populations. As Streiner, Norman & Cairney (2015) highlight, a comprehensive evaluation of a test’s psychometric properties should include not only correlations but also analysis of measurement error, sensitivity, and specificity—particularly when translating a tool for clinical use. From a clinical standpoint, these findings are promising. The test shows strong potential as a reliable and valid measure of auditory temporal resolution, which is relevant in diagnosing auditory processing disorders, assessing hearing aid performance, and evaluating cochlear implant outcomes. However, further research may be needed to establish normative data, evaluate test–retest reliability, and assess predictive validity in diverse clinical populations.

The key limitation of this study is the absence of a clinical reference group, such as individuals with confirmed APD, which restricts our ability to evaluate the diagnostic accuracy of the test in a real-world clinical context. To address this, future research should include a follow-up diagnostic accuracy study involving Thai-speaking participants with formally diagnosed APD. This would allow for the assessment of sensitivity and specificity, essential metrics for determining the clinical utility of the test. Such a study should be designed following established reporting and methodological standards, such as the Standards for Reporting Diagnostic Accuracy Studies (STARD) guidelines (Bossuyt et al., 2015). These guidelines emphasize the importance of using an appropriate reference standard, ensuring blinding of assessors, and providing clear reporting of participant selection, index test procedures, and statistical methods. By following these best practices, future studies can more rigorously determine whether this gap-detection test is a reliable and valid diagnostic tool for identifying APD in Thai-speaking populations. This study presents the first preliminarily normative data for the Thai version of the GIN test in a young adult population, offering an important foundation for future clinical and research applications. The observed associations between gap-detection thresholds and response accuracy support the theoretical validity of the test. However, due to methodological limitations—including a narrow age range, lack of clinical validation, and concerns about statistical and reliability issues—these findings should be interpreted with caution.

While the counterbalanced rotational design does offer a structured way to reduce biases related to the order of tests, there are still various factors to consider that could impact the results, such as fatigue, practice effect and learning transfer. Without proper randomization, the counterbalanced rotational design may fail to equally distribute potential biases or confounding variables across the test sequences. Randomization plays a crucial role in ensuring that each test sequence is equally represented, so that external factors, such as participant characteristics, don’t systematically affect the results.

However, our study’s inclusion criteria were carefully designed to ensure that participants had similar baseline characteristics, minimizing the likelihood that these factors would influence the results. As a result, the differences between participants in terms of key demographic and baseline variables were minimal. This strengthens the validity of our findings by reducing the risk of confounding variables affecting the outcomes.

Nevertheless, despite the well-designed inclusion criteria, it remains a limitation of the study that potential subtle differences in baseline characteristics may still exist. These small differences, though unlikely to significantly impact the results, could still have some effect. Moreover, there could be unmeasured variables or other factors that, although not directly related to the inclusion criteria, may still influence participant responses in ways we have not accounted for. Therefore, while we believe that the impact of these factors is minimal, it is still a limitation that needs to be acknowledged. The uncontrolled confounding variables may have influenced the results. These include subtle differences in stimulus presentation (e.g., headphone fit), cultural or linguistic influences on auditory perception, and individual differences in prior musical training or auditory experience. These factors were not systematically measured or controlled, and may contribute to variability in performance. Further studies are necessary to evaluate the test’s diagnostic performance in Thai speakers with confirmed APD, using rigorous methodological frameworks. Additionally, expanding the sample to include a broader demographic range and assessing test–retest reliability will be essential steps toward establishing the clinical utility of the Thai GIN test.

Overall, our findings contribute to a deeper understanding of how language background influences auditory processing and may inform the development of tailored interventions for individuals with APD. Moreover, our comparison revealed significant differences in auditory processing among Thai, English, Korean, and Portuguese language users. These findings underscore the critical importance of considering linguistic factors when assessing auditory processing abilities, emphasizing the need for country-specific norms in evaluating APD.

Our study contributes to the growing body of literature on auditory processing abilities in diverse populations, highlighting the substantial influence of language on auditory perception. By establishing preliminary normative values for the GIN test in Thai native language users, our findings provide valuable clinical insights for diagnosing and managing APD within this population. Future research should aim to explore additional linguistic and cultural factors that may influence auditory processing, as well as validate the efficacy of country-specific assessment protocols for APD diagnosis and intervention. Such investigations will enhance our understanding of auditory processing across different language groups and may lead to improved diagnostic tools and therapeutic strategies tailored to the unique needs of various populations.

Conclusion

The results revealed that Thai native language users exhibited notable temporal processing abilities, with A.th. values of 5.12 ± 0.81 ms. in the right ear, 5.08 ± 0.81 ms. in the left ear, and 5.04 ± 0.68 ms. when tested bilaterally. The OPC values were found to be 69.94 ± 7.66% in the right ear, 71.37 ± 7.21% in the left ear, and 71.35 ± 6.72% in both ears. Interestingly, while no significant differences were observed in the A.th. between the ears, variations in the OPC responses were noted, suggesting potential asymmetries in auditory processing mechanisms.

Furthermore, our findings highlighted the influence of linguistic factors on auditory processing, as evidenced by the differences in A.th. values between Thai native language users and individuals from other language backgrounds. These results underscore the importance of considering language-specific norms when assessing APD in clinical settings. Establishing such norms is crucial for accurate diagnosis and effective intervention strategies tailored to the linguistic characteristics of the population being evaluated.

Supplemental Information

Supplemental Information 1 Data analysis script

Supplemental Information 2 Raw data

Supplemental Information 3 Data codebook

Additional Information and Declarations

Competing Interests

Author Contributions

Human Ethics

Data Availability

The authors declare there are no competing interests.

Supakarn Jeanbunjongkij conceived and designed the experiments, performed the experiments, analyzed the data, prepared figures and/or tables, authored or reviewed drafts of the article, and approved the final draft.

Pawichaya Suphinnapong performed the experiments, analyzed the data, prepared figures and/or tables, authored or reviewed drafts of the article, and approved the final draft.

Patcharaorn Limkitisupasin performed the experiments, authored or reviewed drafts of the article, and approved the final draft.

Yuttana Roongthumskul conceived and designed the experiments, authored or reviewed drafts of the article, and approved the final draft.

Nattawan Utoomprurkporn conceived and designed the experiments, authored or reviewed drafts of the article, and approved the final draft.

The following information was supplied relating to ethical approvals (i.e., approving body and any reference numbers):

The study received ethical approval from the Research Ethics Review Committee for research participants, Group 1, Chulalongkorn University. COA No.008/2022, Study number 198.1/64.

The following information was supplied regarding data availability:

The data and code are available in the Supplemental Files.

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
