# Peer review of "Auditory processing ability in Thai native speakers with the Gaps-In-Noise (GIN) test"

_PeerJ, doi:10.7717/peerj.20207_

## Round 0.1 · original submission · Major Revisions

All reviewers indicated that the work described in the current version of the manuscript addresses an area of interest and a current deficit in the literature (i.e. understanding temporal processing in native speakers of a tonal language). However, each reviewer has identified issues with the current manuscript which will need to be addressed prior to consideration for publication. in particular, concerns were expressed around the appropriateness of the statistical analysis employed, along with some deviations from the original study protocol. In addition, several points were identified as limitations of the current study with respect to its clinical relevance for native Thai speakers with suspected APD.
In your revised manuscript, please carefully note ad address the points raised by the reviewers, and indicate where in the revisions each amendment is located. Please also note that the revised manuscript will be sent out for re-review prior to a publication decision. While every effort is made to enlist the same reviewers for the revised and original manuscripts, this is not always possible. I hope that you will consider the reviewer comments as constructive feedback, and that the points raised will prove useful to you in revision.

Reviewer 1 ·

Basic reporting

The paper addresses an important area of auditory processing research by establishing normative data for the GIN test specifically for Thai native speakers. The abstract clearly outlines the study's purpose, methodology, and key findings. The researchers have identified a significant knowledge gap, as Thai, being a tonal language, may influence auditory processing differently compared to non-tonal languages. Establishing normative data for Thailand represents a valuable contribution to both audiology research and clinical practice, where language-specific assessment tools for auditory processing disorders are currently lacking. However, upon close examination, there are several methodological and statistical concerns that need addressing to ensure the reliability of these normative data.
Novelty and significance
The study makes several novel contributions to the field:
- It provides the first normative data for the GIN test in Thai native speakers
- It examines potential differences in auditory processing between ears, offering insights into possible asymmetries in auditory processing mechanisms.
- It compares auditory processing abilities between Thai speakers and individuals from other linguistic backgrounds, advancing the understanding of cross-linguistic differences in temporal processing.
However, the narrow age range (18-25 years) limits generalizability, and the absence of clinical validation in Thai populations with suspected APD reduces the immediate clinical impact. To enhance significance, future work should extend these norms to pediatric and older populations and validate diagnostic accuracy in Thai clinical cohorts. This issue should be addressed as a limitation.

Experimental design

Several methodological issues require attention, ranked by importance:
1. Sample size calculation: The authors use the Hajian-Tilaki formula (lines 167-176) targeting specificity rather than precision of normative estimates. This approach is inappropriate for establishing clinical norms. Recommendation: Recalculate sample size based on desired confidence interval width for means or percentile cut-offs that would be used clinically, following guidelines by Jennen-Steinmetz & Wellek, 2005 (https://doi.org/10.1002/sim.2177) on sample sizes for reference intervals.
2. Statistical analysis approach: While paired t-tests and one-sample t-tests were appropriately selected, no normality checks were reported, and there are no corrections for multiple comparisons. As noted by Rothman (1990, doi: 10.1097/00001648-199001000-00010), conducting eight cross-language comparisons without adjustment increases Type I error risk substantially. Recommendation: Perform and report normality tests (e.g., Shapiro-Wilk) for all variables and apply appropriate corrections (Bonferroni or Holm) for multiple comparisons, adjusting significance thresholds accordingly.
3. Test protocol inconsistency: Only three of four available GIN lists were administered (lines 146-153) without an inter-list equivalence analysis, unlike Musiek et al., 2005 (https://doi.org/10.1097/01.aud.0000188069.80699.41), who confirmed list homogeneity. Recommendation: Either conduct and report an equivalence analysis showing that list omission does not bias threshold estimates, or acknowledge this limitation and discuss its potential impact on results.
4. Data transparency: The raw data file lacks sufficient variable labels and list identifiers. Recommendation: Provide a comprehensive data dictionary and analysis script (e.g., R/SPSS syntax) to enhance reproducibility, following FAIR data principles as outlined by Wilkinson et al., 2016 (https://doi.org/10.1038/sdata.2016.18).
The inclusion/exclusion criteria are well defined (lines 118-127), and the testing environment (anechoic chamber) provides appropriate standardized conditions, which are strengths of the methodology.

Validity of the findings

Results
The results presentation needs refinement in these specific areas:
1. Figure 1 shows a statistically significant 1.44% lower OPC in the right ear compared to the left ear (p=0.017). I suggest explicitly discussing the clinical relevance (or lack thereof) of this small difference (<2% absolute) and consider adding an effect size measure.
2. Table 3 reports Thai A.th. values being higher than Portuguese (Δ 0.85 ms, p<0.001) but similar to Korean values. Recommendation: Apply and report a correction for multiple comparisons (e.g., Bonferroni correction would adjust α to 0.00625), then revise interpretations based on corrected significance levels. Only the Portuguese comparison would likely remain significant under this correction.
3. The strong negative correlations between A.th. and OPC values (-0.907 for right ear, -0.839 for left ear, -0.893 for both ears) demonstrate good internal consistency, but lack clinical context. Recommendation: Discuss what these correlations mean for test validity and reliability in clinical settings, referring to established psychometric literature (e.g., Streiner et al., 2015, https://doi.org/10.1093/med/9780199685219.001.0001).
4. While basic descriptive statistics are provided (Table 1), clinical cut-off values aren't established. Recommendation: Calculate and report percentile-based cut-offs (e.g., 5th or 10th percentiles) that would be most relevant for clinical application, following guidelines by ASHA (2005) for auditory processing disorder assessment.

Discussion and Conclusions
The discussion requires several refinements:
1. The authors over-interpret the small OPC asymmetry (lines 243-250) as evidence for "auditory processing mechanisms." As Wasserstein & Lazar (2016, https://doi.org/10.1080/00031305.2016.1154108) advise, avoid over-interpreting small p-values. Recommendation: Revise the statement accordingly.
2. The statement that "linguistic factors appeared to influence the outcomes" (lines 265-269) does not adequately consider methodological differences between studies. Recommendation: Temper this claim by discussing how methodological variations between studies (equipment, testing protocols, sample characteristics) might contribute to observed differences. Consider methodological factors that could account for cross-linguistic variations, as methodological differences rather than language effects alone could explain some observed discrepancies.
3. While suggesting Thai-specific norms for clinical use is reasonable, this lacks validation in APD populations. Recommendation: Explicitly acknowledge this limitation and suggest a follow-up study examining diagnostic accuracy (sensitivity/specificity) in Thai speakers with confirmed APD, following guidelines like those by Bossuyt et al., 2015 (https://doi.org/10.1136/bmj.h5527) for diagnostic accuracy studies.
4. The discussion insufficiently addresses key limitations. Recommendation: Add a dedicated limitations section discussing: (a) narrow age range, (b) lack of clinical validation, (c) statistical methodology issues, and (d) test reliability concerns given the modified protocol.
The conclusions should be more measured given these methodological concerns, while still acknowledging the value of establishing first Thai GIN norms.

Additional comments

Writing quality and clarity
The paper requires linguistic refinements:
1. Terminology error: "OTC" is used instead of "OPC" in the abstract (line 100 and line 184). Recommendation: Correct this inconsistency and perform a full-text search for other similar mistakes.
2. Protocol description: The GIN test administration description (lines 146-158) lacks procedural details. Recommendation: Expand to include specific instructions given to participants, number of practice trials, interval between trials, and exact equipment specifications.

Figures and Tables
Only minor improvements to figures and tables are needed:
1. Table 3 inconsistently reports statistics, with means ± SD for Korean data but only means for other languages. Recommendation: Either obtain and report SD values for all comparative data, or add footnotes explicitly stating "SD not reported in original study" where applicable.
I confirm that the figures have not been inappropriately manipulated based on visual inspection.

Overall, this paper makes a valuable contribution by establishing the first Thai-language normative data for the GIN test. However, several methodological and statistical issues need to be addressed before publication.
Addressing these issues would strengthen the manuscript and enhance its utility as a reliable clinical reference for Thai-speaking populations. The core contribution is valuable, but the methodological foundation needs reinforcement.

·

Basic reporting

Language and Style:
The manuscript is written in generally professional academic English. However, some sections, particularly lines 58-66 in the introduction, are wordy and could benefit from more concise phrasing to enhance clarity.

Background and Literature:
The introduction provides sufficient background information, and the relevant literature is appropriately cited. References are recent and pertinent.

Structure:
The structure of the manuscript adheres to PeerJ standards and the norms of the discipline. All essential sections (Introduction, Methods, Results, Discussion, Conclusion) are properly presented.

Figures and Tables:
Figures and tables are relevant; however, Figure 1 and Figure 3 require clearer labeling and captions. It should be explicitly stated which groups are being compared in these figures.

Experimental design

Originality:
The study addresses a relevant gap by providing normative data for the GIN test in native Thai speakers, which is a meaningful contribution.

Research Question:
The research question is clearly stated and relevant. The rationale for the study is adequately described.

Ethical Standards:
Ethical approval is obtained and explicitly stated. Procedures for human participants appear ethically sound.

Methodological Detail:
The procedures are described with sufficient detail. However, the justification for using parametric statistical tests (e.g., paired t-test) given the relatively small sample size (n=52) is missing. It should be confirmed whether assumptions of normality were tested.

Limitations:
The age range of participants (18-25) is narrow. Therefore, claiming to establish "normative data" should be moderated to reflect this limitation. The limited generalizability of findings to other age groups needs to be more explicitly discussed.

Validity of the findings

Data Robustness:
The statistical analyses are generally appropriate. The use of Pearson correlations and t-tests is acceptable, but as mentioned, normality assumptions must be addressed.

Strength of Conclusions:
Conclusions are well aligned with the research questions. However, when comparing findings across different language groups (English, Polish, Korean, Portuguese), potential confounding variables (such as differences in stimulus presentation, cultural factors, or participants' musical training) are not discussed, which weakens the cross-linguistic comparisons.

Limitations Handling:
The study does not sufficiently discuss its inherent limitations related to sample size, age homogeneity, and the potential influence of uncontrolled variables.

Additional comments

Improve Figure and Table Captions:
Figures, particularly Figure 1 and Figure 3, should provide more descriptive captions indicating exactly which groups are being compared.

Clarify Statistical Methods:
It should be explicitly stated whether normality was tested before applying parametric tests. If normality was not confirmed, non-parametric alternatives (e.g., Wilcoxon signed-rank test) should be considered.

Moderate the Claim of Providing Normative Data:
Since the sample is limited to young adults (18-25 years), it would be more appropriate to refer to the findings as "preliminary normative data for young adults" rather than for the general Thai-speaking population.

Expand the Discussion on Linguistic Influence:
The influence of linguistic factors on auditory processing should be discussed in greater depth, acknowledging that cross-language comparisons may be influenced by cultural, educational, and methodological differences.

Reviewer 3 ·

Basic reporting

The manuscript is clear and unambiguous, but could be improved. The intro & background are appropriate. Literature cited is relevant. Figures are relevant and require minor corrections (labelling). Figures (appearance) can be improved.

Experimental design

Research question well defined, relevant & meaningful. It is stated how the research fills an identified knowledge gap. Methods are described with sufficient detail & information to replicate.

Validity of the findings

Conclusions are well stated, linked to the original research question

Additional comments

Abstract
line 47: ‘study revealed consistent temporal processing abilities among native Thai speakers’ – here, what does consistent mean?

Introduction
Line 58-59: Consider adding a citation for the definition.
Lone 80: (Paulovicks 2008) is an incorrect citation
Line 89-90: add appropriate citations to support the claim
Line 193: APT – what is this?
Line 210-211: The first statement can be merged after line 200. But why show both the table and the figure for the same information?

Objective 1 was to establish normative.
Normative score or cutoff score not reported.

Objective 2 was to compare scores of right, left, and both ears.
The tests used to compare scores across ear conditions are not reported.

Objective 3 was to compare the mean (not normative) values of the GIN test between Thai native speakers and individuals from other linguistic backgrounds.
It requires correction

Table 1. The range for the left ear and both ears is 4 to 6, but in Figure 3 range is 3.0 to 6.0 for the left ear and both ears

---

## Round 0.2 · accepted · Accept

I look forward to seeing this work in final published form.